# Association of the awareness of the epidemic, mental health status with mobile phone screen use time in Chinese college students during COVID-19 isolation and control

**Min Fang[1], TengChi Ma[1]◉, HongHong Li[1]◉, Tuo Han[2]◉, JiaJia Wang[1]◉, ZhiLe Li[1]◉, Jing Zhou[3]***

**1** Yan'an University, Yan'an, China, **2** Department of Cardiology, The Second Affiliated Hospital of Xi'an Jiaotong University, Xi'an, China, **3** Department of Cardiovascular Medicine, Affiliated Hospital of Yan'an University, Yan'an, China

◉ These authors contributed equally to this work.
* 25891082@qq.com

**Data Availability Statement:** The data supporting our findings is included within this article, and for more details, see the Supporting information files.

## Abstract

### Objective

This study aimed to investigate the awareness of the epidemic among college students and their mental health as well as to explore the association between their awareness of the epidemic mental health and the daily mobile phone screen use time, in order to provide guidance for the publicity of school epidemic prevention and control knowledge and the psychological counseling of students.

### Methods

A cross-sectional design was employed among 780 college students, The Pandemic Fatigue Questionnaire, epidemic prevention and control knowledge and the mental health Scale were used to collect data through an online survey.

### Results

1. Awareness rate of the transmission routes and protective measures of COVID-19 among college students is higher when the daily mobile screen use time is 3–7 hours. 2. 21.79% of the 780 college students felt stressed; 24.87% felt anxious; 19.23% showed depression. 3. The scores of each subscale in the daily mobile phone screen use time of 3–7 hours and more than 7 hours were higher, and the scores of each subscale in the group of more than 7 hours were the highest. Further correlation analysis found that the time spent on mobile phone screens was positively correlated with stress, anxiety, and depression scores (r = 0.155, 0.180, 0.182, P<0.01).

### Conclusion

During the COVID-19 isolation and control period, college students with different mobile screen usage time have different understandings of the epidemic. Long-term mobile screen

**Funding:** Special research project on epidemic prevention and control and economic and social development of Yan'an University (YCX2022011), Min Fang; Yan'an University Epidemic Prevention and Control Emergency Scientific Research Project (ydfk006), Jing Zhou. The funders had no role in study design, data collection and analysis, decision to publish, or preparation of the manuscript.

**Competing interests:** The authors have declared that no competing interests exist.

use is related to the occurrence of psychological problems such as stress, anxiety, and depression. Therefore, education departments and schools should pay attention to college students' mobile phone use time to reduce the occurrence of bad psychological state of students.

## 1. Introduction

The novel coronavirus outbreak outbreak from Wuhan, in China at the end of 2019 and spread rapidly around the world [1], which known as coronavirus disease 2019 (COVID-19) and has caused Mass Mortalities. Therefore, the World Health Organization considers COVID-19 to be the most serious public health emergency after SARS. We immediately took isolation and control measures to control the spread of the virus. However, being restricted from activities for a long time has caused great changes in the psychology of many people. [2, 3], it is easy to produce negative emotions such as pressure and anxiety, especially for college students who have extensive interpersonal relationship and heavy learning tasks, which is a huge challenge [4].

A recent study showed that the detection rate of anxiety and depression was significantly increased among ordinary Chinese adults during the COVID-19 Pandemic [5]. College students often live in groups, but the sudden isolation and control made them show varying degrees of negative emotions [6]. In order to prevent the spread of the virus on the university campus, the school prohibits students from walking around the campus and conducting online learning in the dormitory. College students lack interpersonal relationships, so they are more prone to psychological stress, anxiety and even depression [7]. The time students spend on electronic devices is significantly longer during the quarantine period; Studies have shown that isolation will increase the time spent on the Internet and increase the generation of negative emotions. studies have found that quarantine increases internet usage and negative emotions. At the same time, the mental damage caused by the sudden global public health event may become a long-term health problem [8]. In the past, Many studies focus on the psychological problems of college students during home isolation, and parents will take care of them. However, They have less concern from family and friends, are more likely to develop psychological problems during dorm isolation. The purpose of this study is to investigate college students' understanding of epidemic prevention and control knowledge and their mental health status, and to provide scientific guidance for schools to respond to epidemic prevention and control work and manage students.

## 2. Objectives and methods

### 2.1. Objective

Participants were recruited from the student population of Yan'an University. This university closed all their campuses on January 9, 2022, and held all its classes virtually in response to the COVID-19 pandemic. We use the "Questionnaire Star" platform to make online questionnaires, and collect data by forwarding the class's WeChat group. In this study, the questionnaires with missing items or filling in more than 120s were excluded, and the final valid data was 780 copies. This study was approved by the Ethics Review Board of Yan'an University (approval number: YAU-G202207002).

## 2.2. Study design

**2.2.1. Demographics.** Demographics included age, gender, profession, hukou, family relationship, relationship, single parent, and daily mobile screen time.

**2.2.2. Epidemic awareness survey.** Transmission route: Do you think the transmission route of the new coronavirus is: aerosol transmission, air transmission, contact transmission, fecal-oral transmission; prevention and control measures: Do you think the best prevention and control measures to deal with the epidemic are: home isolation, N95 masks, ordinary Masks, hand sanitizers.

**2.2.3. DASS-21 scale.** The DASS-21 scale we used was revised by Henry et al in 2005 to assess the state and severity of stress, anxiety and depression in college students [9]. There are 21 questions in total, and each question is scored between 0 and 3 points. "0" means "never", "1" means "sometimes", "2" means "often", and "3" means "always". Yes", the total score of each subscale is multiplied by 2 to obtain the final score, which is divided into 5 grades according to the score, namely normal, mild, moderate, severe and extremely severe. The scale has good reliability and validity.

## 2.3. Statistical analysis

SPSS26.0 statistical software package was used for statistical processing; measurement data were expressed as mean ± standard deviation or median and interquartile range, t test was used between two groups, one-way ANOVA was used for multiple groups, and count data were used Percentage, representation, and correlation between variables were analyzed by Spearman correlation, Significance level of $p < 0.05$ was fixed for the analysis.

## 3. Results

### 3.1. Demographic description

A total of 780 data were collected, of which 478 were female respondents (61.3%). 440 (56.4%) of participants were younger than 20 years old. The sample included medical students (n = 154, 19.7%) and non-medical students (n = 626, 80.3%). Among them, urban household registration (n = 240, 30.8%), rural household registration (n = 540, 69.2%), more than 2 siblings (n = 136, 17.4%), good family relationship (n = 466, 59.7%), Single-parent family (n = 48, 6.2%), currently in a relationship (n = 218, 28.0%), time spent on mobile phone screen < 3h (n = 154, 19.7%), 3-7h (n = 434, 55.6%)), >7h (n = 192, 24.6%).

### 3.2. The cognition of college students with different mobile phone screen usage time on the transmission route and protective measures of the epidemic

Awareness rate of COVID-19 transmission routes and prevention and control measures for college students with different mobile phone screen usage time: Awareness rate of college students with different mobile phone screen usage time <3 hours: aerosol transmission (15.6%), air transmission (45.5%), contact transmission (37.7%), fecal-oral transmission (1.3%); home isolation (48.1%), N95 masks (19.5%), ordinary masks (32.5%), hand washing and disinfection (0.0%); mobile phone screen usage time varies by 3–7 hours, college students know Rates: aerosol transmission (20.3%), air transmission (39.2%), contact transmission (40.1%), fecal-oral transmission (0.5%), home isolation (54.8%), N95 masks (23.5%), ordinary masks (18.0%)) Handwashing and disinfection (3.7%); Awareness rate of college students with different mobile screen usage time > 7 hours: Transmission route: aerosol transmission (18.8%), air transmission (45.8%), contact transmission (33.3%), fecal-oral transmission (2.1%); protective measures:

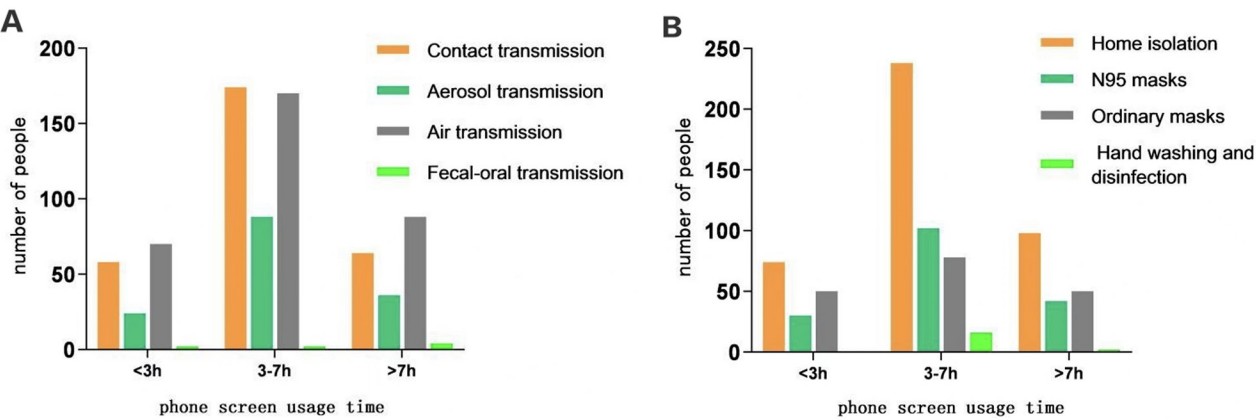

**Fig 1. Comparison of college students' cognition of the epidemic situation by different mobile phone screen usage time.** (a)College students' knowledge of the transmission route of the epidemic at different time online; (b) College students' knowledge of epidemic prevention and control measures in different time online.

protective measures: home isolation (51.0%), N95 masks (21.9%) ordinary masks (26.0%) hand washing and disinfection (1.0%). When the mobile phone screen usage time was 3–7 hours, the awareness rate of the transmission route and protective measures of COVID-19 was higher among college students, and there was no difference in the awareness of the transmission route of COVID-19 among the three groups ($\chi2$ = 8.422, P = 0.209). Statistically significant, the difference in protective measures ($\chi2$ = 22.041, P = 0.001) was statistically significant (Fig 1).

### 3.3. Severity distribution of different negative emotions in college students' mobile phone screen use time

According to DASS-21 subscale scores and severity grading standards, the prevalence rate of stress among 780 college students was 21.8%, among which the prevalence rate of mobile phone screen use time >7h was the highest, 42 were mild (21.9%), moderate and moderate. More than 14 people (7.3%); the prevalence of anxiety is 17.69%, of which the prevalence of mobile phone screen use time > 7 hours is the highest, 12 people (6.3%) are mild, and 50 people (20.6%) are moderate and above; The prevalence of depression was 24.87%, with the highest prevalence of mobile phone screen use for >7 hours, 26 patients (13.5%) were mild, and 34 (17.7%) were moderate and above (Fig 2).

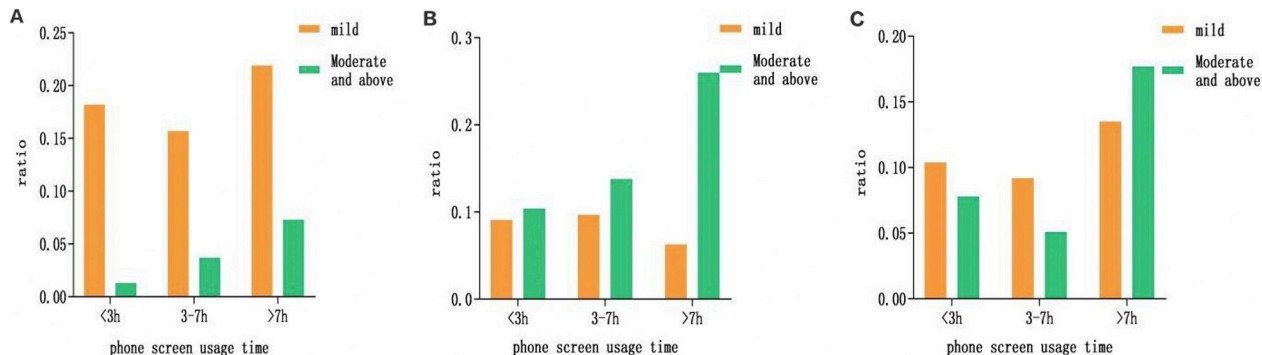

**Fig 2. Comparison of the severity of negative emotions in different college students' mobile phone screen use time.** (a)Comparison of different stress severity of college students' mobile phone screen use time; (b)Comparison of anxiety severity among college students with different mobile phone screen usage time; (c)Comparison of depression severity among college students with different mobile phone screen usage time.

**Table 1. Comparison of DASS-21 subscale scores of college students with different mobile phone screen usage time(M;Q1,Q3).**

| Daily phone screen time | stress score | Anxiety Score | Depression Score |
|---|---|---|---|
| 3h | 4.0 (0.0, 10.0) | 2.0 (0.0, 6.0) | 2.0 (0.0, 6.5) |
| 3-7h | 4.0 (2.0, 10.0) | 2.0 (0.0, 6.0) | 2.0 (0.0, 6.0) |
| >7h | 8.0 (2.0, 12.0) | 4.0 (2.0, 10.0) | 4.0 (2.0, 10.0) |
| H | 23.560 | 29.851 | 39.084 |
| P | <0.001 | <0.001 | <0.001 |

### 3.4. Comparison of scores of DASS-21 subscales of college students with different mobile phone screen usage time

The mobile phone screen usage time, compared with the mobile phone screen usage time of <3h group, the use time of 3-7h group and >7h group, the scores of each subscale are higher, compared with the mobile phone screen usage time of 3-7h group, the use time The scores of each subscale in the >7h group were higher, and the difference was statistically significant (Table 1).

### 3.5. Correlation analysis between mobile phone screen use time and negative emotions of college students

According to the correlation analysis results, it is shown that the time spent on the screen of college students' mobile phone is positively correlated with stress, anxiety and depression (r = 0.179, 0.160, 0.215, P<0.001), that is, the longer the screen time of college students is, the greater the risk of stress, anxiety and depression. the higher the score (Fig 3).

## 4. Discussion

The outbreak of novel coronavirus affects the public's work, study and daily life [10], especially for college students. For college students, they can only learn through the Internet, but it cannot completely replace traditional classroom learning in terms of certain skills practice [11]. Secondly, college students' awareness of the COVID-19 virus also affects their mentality [12]. A study on the effectiveness of public health measures in reducing the spread of the COVID-19 virus found that some personal protection and social measures, including hand washing, wearing a mask and physical distancing, reduce the incidence of COVID-19. The COVID-19

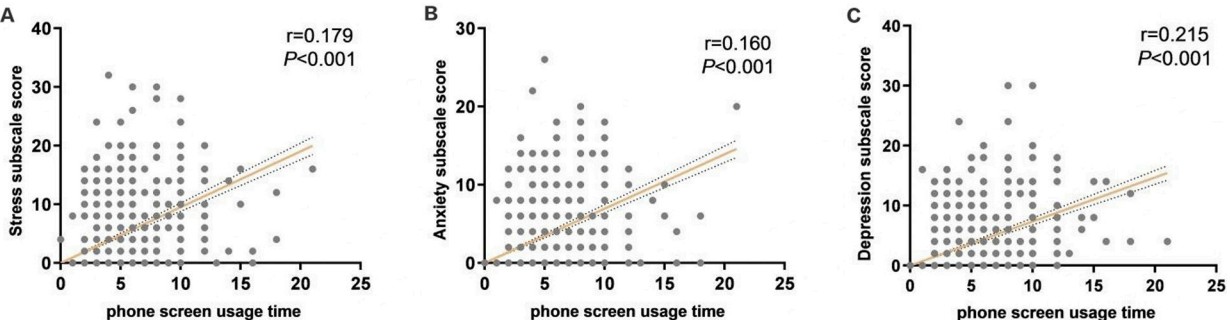

**Fig 3. Correlation analysis between mobile phone screen use time and negative emotions of college students.** Correlation analysis between mobile phone screen use time and stress of college students; (b)Correlation analysis of mobile phone screen time and anxiety among college students; (c)Correlation analysis of mobile phone screen use time and depression in college students.

virus is mainly spread through infectious droplets secreted by an infected person's respiratory system, and the virus is spread through the respiratory droplets of an infected person when they sneeze, cough or talk [13]. COVID-19 can also be transmitted through [14]: (a) expelled droplets may also linger in the air and infect individuals who come into contact with them in enclosed spaces, (b) direct physical contact with infected individuals, touching a contaminated surface or object, touching the mouth, eyes or nose with unwashed and contaminated hands, (c) a very small fraction of the virus is transmitted through feces. A study on the effectiveness of public health measures in reducing the incidence of covid-19, SARS [15] Several personal protection and social measures, including hand washing, wearing a mask, and physical distancing, can reduce the incidence of COVID-19. We conducted a survey on college students' understanding of the transmission routes and protective measures of the epidemic. The results showed that college students who use mobile phone screens for 3–7 hours believe that the transmission routes of COVID-19 are aerosol transmission, contact transmission and air transmission, and personal prevention and control measures are the most important. Effective should be home isolation, N95 masks, The most effective control measures should be home isolation and N95 masks. Therefore, when the mobile phone screen is used for 3–7 hours, students are more aware of the transmission routes of the COVID-19 virus and protective measures, which is helpful for them to prevent the COVID-19 virus.

In the wake of the COVID-19 outbreak, most countries have implemented various measures to reduce the spread of the virus, such as social distancing, lockdowns and self-isolation, all of which can adversely affect the public's mental health. As a result, psychological issues have become a hot topic of global concern, as during the SARS virus outbreak, studies have found that isolation has a psychological impact on people in Toronto, Canada [16] Similarly, at the initial stage of the COVID-19 outbreak, a study in China found that the prevalence of stress, anxiety, and depression among Chinese were 8.1%, 28.8%, and 16.5%, respectively [17]. For college students, they are in various stages of physical and psychological growth. Will the outbreak of COVID-19 and isolation and control measures have an impact on their mental health? A study surveyed American college students and found that 71.26% of college students showed Out of Stress and Anxiety [18]. The survey of this study found that 170 (21.79%) of the 780 college students felt stressed after isolation; 194 (24.87%) felt anxious; 150 (19.23%) showed depression, with moderate and above being 4.10%, 16.15%, and 8.72%, respectively. During the COVID-19 pandemic, people have been asked to stay at home or in dormitories to slow the spread of the virus, and outdoor social activities have been drastically reduced. An online survey of the general Chinese population showed that during the pandemic, 46.8% of participants reported increased reliance on internet use, while 16.6% reported longer duration of internet use [19]. Another study in Japan showed that [20], due to home isolation, excessive gaming or Internet users reported that With increasing [20], although home isolation and Internet use can facilitate prevention and further infection control, it can lead to excessive use of the Internet and gaming by students. A study showed an increase in Internet use among children and adolescents during the COVID-19 epidemic, including the frequency and duration of recreational Internet use, and the frequency of staying up late [21, 22]. During the epidemic, as college students were confined to dormitories and had to stay in dormitories for online learning and necessary social activities on the Internet, this naturally prolonged their use of the Internet. In addition, the use of the Internet is a strategy for coping with stressful living conditions, and moderate leisure and entertainment can relieve stress [23]. This study found that the longer the time spent on the Internet, the higher the score of the DASS-21 scale, and the higher the incidence of stress, anxiety, and depression in the 3–7 hours of Internet time. Correlation analysis was used to analyze these influencing factors, and the results found that the time spent on mobile phone screens was positively correlated with the occurrence of

stress, anxiety and depression. Therefore, some guidance should be given to students using electronic devices to surf the Internet.

All in all, for special groups such as college students, they have a certain understanding of the control and prevention of COVID-19 disease in the process of using electronic devices, but due to the lack of self-control of students, it is easy to use the Internet for a long time for games, shopping, etc. Chat, etc., have a certain degree of impact on their psychology. Therefore, when guiding the psychological problems of college students, we should not only pay attention to traditional influencing factors such as age, gender, etc., but also pay attention to the time of mobile phone use, and provide corresponding psychological support to students with moderate to severe psychological problems.

## Supporting information

**S1 Data.**
(XLSX)

## Author Contributions

**Methodology:** Tuo Han.

**Project administration:** TengChi Ma.

**Writing – original draft:** Min Fang, Jing Zhou.

**Writing – review & editing:** HongHong Li, JiaJia Wang, ZhiLe Li.

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
