## [Decision Letter · Decision Letter 0]

13 Sep 2022

PGPH-D-22-01015

Research on the correlation between mobile phone screen use time and epidemic cognition and mental health status of college students during COVID-19 isolation and control

Dear Dr. Zhou,

Thank you for submitting your manuscript to PLOS Global Public Health. After careful consideration, we feel that it has merit but does not fully meet PLOS Global Public Health’s publication criteria as it currently stands. Therefore, we invite you to submit a revised version of the manuscript that addresses the points raised during the review process.

We look forward to receiving your revised manuscript.

Kind regards,

Muhammad Fawad Rasool

Academic Editor

Journal Requirements:

1. In the online submission form, you indicated that [Insert text from online submission form here]. All PLOS journals now require all data underlying the findings described in their manuscript to be freely available to other researchers, either 1. In a public repository, 2. Within the manuscript itself, or 3. Uploaded as supplementary information.

Additional Editor Comments (if provided):

Reviewers' comments:

Reviewer's Responses to Questions

**Comments to the Author**

1. Does this manuscript meet PLOS Global Public Health’s publication criteria? Is the manuscript technically sound, and do the data support the conclusions? The manuscript must describe methodologically and ethically rigorous research with conclusions that are appropriately drawn based on the data presented.

Reviewer #1: No

Reviewer #2: Yes

2. Has the statistical analysis been performed appropriately and rigorously?

Reviewer #1: No

Reviewer #2: No

3. Have the authors made all data underlying the findings in their manuscript fully available (please refer to the Data Availability Statement at the start of the manuscript PDF file)?

Reviewer #1: Yes

Reviewer #2: Yes

4. Is the manuscript presented in an intelligible fashion and written in standard English?

Reviewer #1: No

Reviewer #2: No

5. Review Comments to the Author

Reviewer #1: Comments

1.Manuscript has a date modified printed on it April 2021 while date of survey claimed in the script is January 9-12 2022 (need to checked)

2.The questions regarding transmission route and protective measures supposed to be answered with multiple options, but authors reported them as independent options with no overlapping response. This need clarity, the method through which the questions were put in front of respondents.

3.Percentages reported in 2.3 are presented with mixed decimal places. Few (prevalence of stress 21.79%, mild (21.9%)). The percentages must be presented uniformly, I believe single decimal place is enough.

4.The section 2.4, Table 1 expresses the scores for stress, anxiety and depression by using mean±sd. I wonder, whether the normality of data was tested?, as very high standard deviations suggest that data was deviating from normality. If so, in this case the data is better to be presented by median and interquartile range i.e. median (Q1 – Q3). The test for comparison will also change (Kruskal Wallis test is suggested) also need to have a post hoc analysis (Mann Whitney U if Kruskal Wallis and Tukey’s or Tamhane’s in case normality is in the data and ANOVA is applicable)

5.The scatter diagram (section 2.5, Figure 3) is presenting relation between time on mobile per day and DASS 21 score (sub items). It is interesting to observe that many scores go beyond 21(the maximum possible score through 7 items, scored between 0-3). Also the time spent per day exceeds 20 hours (beyond reality but still possible). The author need to recheck their score calculated for stress, anxiety and depression as none can exceed 21.

Redo the analysis

Reviewer #2: The topic is very well selected and has a greater impact of the society, however regarding manuscript i would recommend few changes.

abstract need be revised again and should be simple and have clear meanings.

Graphs are not very clear, I think it should be improved more.

There seems to a confusion regarding the results as the authors says that long mobile screen users know more about the transmission and prevention of SARS virus but at the same time data indicates more anxiety and depression. I think a clear statement should be included regarding this.

The English of the manuscript is weak and it need to be improved.

6. PLOS authors have the option to publish the peer review history of their article (what does this mean?). If published, this will include your full peer review and any attached files.

**Do you want your identity to be public for this peer review?** For information about this choice, including consent withdrawal, please see our Privacy Policy.

Reviewer #1: **Yes: **Muhammad Aasim

Reviewer #2: No

---

## [Decision Letter · Decision Letter 1]

3 Jan 2023

Association of the awareness of the epidemic, mental health status with mobile phone screen use time in Chinese college students during COVID-19 isolation and control

PGPH-D-22-01015R1

Dear 25891082@qq.com Zhou,

We are pleased to inform you that your manuscript 'Association of the awareness of the epidemic, mental health status with mobile phone screen use time in Chinese college students during COVID-19 isolation and control' has been provisionally accepted for publication in PLOS Global Public Health.

Best regards,

Muhammad Fawad Rasool

Academic Editor

Reviewer Comments (if any, and for reference):

Reviewer's Responses to Questions

**Comments to the Author**

1. If the authors have adequately addressed your comments raised in a previous round of review and you feel that this manuscript is now acceptable for publication, you may indicate that here to bypass the “Comments to the Author” section, enter your conflict of interest statement in the “Confidential to Editor” section, and submit your "Accept" recommendation.

Reviewer #1: All comments have been addressed

2. Does this manuscript meet PLOS Global Public Health’s publication criteria? Is the manuscript technically sound, and do the data support the conclusions? The manuscript must describe methodologically and ethically rigorous research with conclusions that are appropriately drawn based on the data presented.

Reviewer #1: Yes

3. Has the statistical analysis been performed appropriately and rigorously?

Reviewer #1: Yes

4. Have the authors made all data underlying the findings in their manuscript fully available (please refer to the Data Availability Statement at the start of the manuscript PDF file)?

Reviewer #1: Yes

5. Is the manuscript presented in an intelligible fashion and written in standard English?

Reviewer #1: (No Response)

6. Review Comments to the Author

Reviewer #1: I am satisfied with the response of authors on the given advise.

1. They have checked the normality of data and replaced ANOVA by Kruskal Wallis test. The Only mistake is they forgot to replace the "F" written in table with "Chi-square" after replacing ANOVA with Kruskal Wallis test. need to change that.

2. Cleared the issue on scale by placing table 1 presenting the range and method of scale explaining the scores were multiplied by 2.

3 . still there are few values displayed with double decimal places, but overall its good.

This is acceptable

7. PLOS authors have the option to publish the peer review history of their article (what does this mean?). If published, this will include your full peer review and any attached files.

**Do you want your identity to be public for this peer review?** For information about this choice, including consent withdrawal, please see our Privacy Policy.

Reviewer #1: **Yes: **Muhammad Aasim

<quillbot-extension-portal></quillbot-extension-portal>